# Associations between white matter integrity and postural control in adults with traumatic brain injury

Cris Zampieri[1]◎*, Jacob B. Leary[1]◎, Pashtun Shahim[1], Diane Damiano[1], Pei-Shu Ho[1], Dzung L. Pham[2], Leighton Chan[1]

**1** Rehabilitation Medicine Department, National Institutes of Health, Bethesda, Maryland, United States of America, **2** Center for Neuroscience and Regenerative Medicine, The Henry Jackson Foundation for the Advancement of Military Medicine, Bethesda, Maryland, United States of America

◎ These authors contributed equally to this work.
* zampierigallac@cc.nih.gov

**Data Availability Statement:** This study is still on-going, therefore data is not available yet. It will only be available once the study is closed and the data is locked, and that may take several years. It will be

## Abstract

Abnormalities of postural sway have been extensively reported in traumatic brain injury (TBI). However, the underlying neural correlates of balance disturbances in TBI remain to be elucidated. Studies in children with TBI have reported associations between the Sensory Organization Test (SOT) and measures of white matter (WM) integrity with diffusion tensor imaging (DTI) in brain areas responsible for multisensory integration. This study seeks to replicate those associations in adults as well as explore relationships between DTI and the Limits of Stability (LOS) Test. Fifty-six participants (43±17 years old) with a history of TBI were tested 30 days to 5 years post-TBI. This study confirmed results in children for associations between the SOT and the medial lemniscus as well as middle cerebellar peduncle, and revealed additional associations with the posterior thalamic radiation. Additionally, this study found significant correlations between abnormal LOS scores and impaired WM integrity in the cingulum, corpus callosum, corticopontine and corticospinal tracts, fronto-occipital fasciculi, longitudinal fasciculi, medial lemniscus, optic tracts and thalamic radiations. Our findings indicate the involvement of a broad range of WM tracts in the control of posture, and demonstrate the impact of TBI on balance via disruptions to WM integrity.

## Introduction

Complaints of balance problems are common following traumatic brain injury (TBI) and may persist for months to years [1,2]. Objectively, these perceived deficits can be detected with a force plate that measures body sway during standing balance tasks. An advantage of force plate-based measures is that they significantly reduce examiner bias, which is not the case for many clinical balance assessment tools [3]. Postural sway abnormalities have frequently been reported in TBI [4–6], and shown to be associated with subjective balance complaints [7,8]. However, their underlying neural correlates remain to be elucidated.

One of the most frequently used force plate-based assessments in TBI is the Sensory Organization Test (SOT) [9], which quantifies abnormalities in the use of sensory information (e.g.

deposited into the Federal Interagency Traumatic Brain Injury Research (FITBIR) https://fitbir.nih.gov/.

**Funding:** This research was supported by the Intramural Research Program of the National Institutes of Health (Protocol #10-CC-0118), and by the Center for Neuroscience and Regenerative Medicine. The funders had no role in study design, data collection and analysis, decision to publish, or preparation of the manuscript.

**Competing interests:** The authors have declared that no competing interests exist.

vestibular, proprioceptive and visual inputs) for postural control. Several studies have reported vestibular and visual deficits on the SOT post-TBI [4,6,8,10]. However, balance assessment in TBI should not rely exclusively on the SOT, as postural control is a complex multicomponent skill [3]. In fact, research has shown that the Limits of Stability (LOS) test, another force plate-based measure, seems to have stronger correlation with subjective balance complaints, uncovering more balance abnormalities in TBI than the SOT [11]. While the SOT is a static balance test, the LOS test quantifies one's ability to intentionally lean their body to the edges of their base of support [9]. Reliability of the LOS in TBI has been well established for over 10 years [12], however only recently has the LOS been recommended to complement balance assessment in this population [11].

There are notable differences between the SOT and the LOS tests: 1) while the SOT measures quiet stance, the LOS requires participants to actively move their body over a stable base; 2) while there is no visual feedback in the SOT, a computer screen provides a target and real-time tracking of the movements of one's center of gravity in the LOS; and 3) the LOS seems to involve cognitive abilities beyond those required during the SOT (e.g. visual information processing, auditory attention, and motor learning) [9]. Therefore, the underlying neural correlates of each test may differ to some degree while also overlapping in some areas. This has yet to be investigated as no studies have directly compared the brain structures correlated with each test.

TBI commonly results in axonal injury due to rotational or acceleration/deceleration-induced shearing forces [13,14]. While this type of injury is undetectable with conventional brain magnetic resonance imaging (MRI), diffusion tensor imaging (DTI) has proven to be highly sensitive in quantifying white matter (WM) abnormalities [15–17]. To our knowledge, research correlating DTI and balance abnormalities in individuals with TBI is still limited, with studies thus far including only pediatric patients with TBI [18,19]. Overall, measures of poor WM integrity of several brain tracts have been significantly correlated with poor balance performance, on clinical balance scales and the SOT. More specifically, lower fractional anisotropy (FA) in the corticospinal tract was significantly associated with lower scores on the Movement Assessment Battery for Children, which suggests impaired balance among other motor deficits [19]. Additionally, reduced FA in the medial lemniscus, internal capsule, cerebellum and cerebellar peduncles correlated with abnormal SOT results [18]. Given these studies were conducted in children, there remains a need for investigation in adults with TBI. Moreover, as the control of posture has multiple components [3] and no single test can fully characterize balance impairments, other specific and objective (e.g. force plate-based) balance assessments, such as the LOS, should be explored along with the SOT.

Our study had three aims. First we attempted to replicate prior research [18] by assessing whether the associations between the SOT and DTI WM abnormalities reported in children with TBI were similar in adults with a history of TBI. Second, we explored potential associations between performance on the LOS test and DTI WM abnormalities in our group of adults with a history of TBI. Third, we investigated relationships between the SOT and LOS balance tests themselves. Identifying these potential associations may assist in better understanding the brain mechanisms underlying balance control and how they may be impaired in adults with TBI.

## Materials and methods

### Participants

This is a cross-sectional analysis that included adult participants from a larger prospective natural history study of TBI. The study was approved by the National Institutes of Health Institutional Review Board and was performed in accordance with the 1964 Declaration of Helsinki and its later amendments. All participants gave written and informed consent before enrolling.

Consent process, data collection and analysis were conducted at the Rehabilitation Medicine Department of the National Institutes of Health Clinical Center. All participants had to be at least 18 years old and had sustained a non-penetrating TBI within the past five years. Exclusion criteria were contraindication to MRI scanning, inability to read or speak sufficient English to complete clinical phenotyping assessments, and any medical or psychological instability that would preclude the ability to complete the study assessments. All participants were seen by a physician, who performed a history and physical and reviewed medical records where available for verification of injury details. The diagnosis of TBI was based on the Department of Veterans Affairs and Department of Defense TBI Severity Rating Scale [20]. Participants were seen either across multiple visits at 30 days, 90 days, 180 days, and one-year post-injury and annually thereafter for up to five years, or for one cross-sectional visit within five years of injury if the participant's injury had occurred greater than one year prior to study enrolment, or if they could not commit to longitudinal visits. Participants were enrolled between August 2011 and January 2019. Data from each participant's first visit was included in this analysis. A longitudinal analysis will be presented in a future publication as this study is still on-going. Assessment was multidisciplinary; however, for the purposes of the present study, we have focused on the neuroimaging and balance evaluations.

To control for factors that could potentially affect our participants' balance scores, we further excluded those with: non-TBI neurologic disorders; lower-extremity orthopaedic surgery within 6 months; symptomatic and/or untreated musculoskeletal conditions in the lower extremities or spine; and uncorrected visual disturbances or uncontrolled central autonomic dysfunction. Participants with hearing impairment were also excluded because the LOS test utilizes an audible cue for trial initiation. All data was deidentified as each participant was attributed a study ID and authors had no access to information that could identify individual participants during or after data collection.

## Balance testing

A NeuroCom SMART Balance Master System (previously Natus Medical Inc, Seattle, WA) was used for the SOT and LOS tests. The SOT is a test of quiet stance that includes six conditions: (1) eyes open, no sway reference; (2) eyes closed, no sway reference; (3) eyes open, visual/surround sway reference; (4) eyes open, support surface sway reference; (5) eyes closed, support surface sway reference; and (6) eyes open, support surface and visual/surround sway reference. Sway reference refers to the displacement of the platform and/or the visual surround in the sagittal plane in response to the sway of the participant. Scores go from 0 (representing a fall) to 100 (representing perfect balance). Three 20-second trials are completed per condition and averaged. SOT metrics are based on an equilibrium score (average of 3 trials), which compares the patient's maximal anteroposterior sway during each trial to the theoretical sway stability limit of 12.5 degrees. High equilibrium scores are indicative of good postural stability while low scores indicate poor stability, and a score of 0 indicates a fall. Fig 1 shows an illustration of the test and representative data for two participants with distinct performances.

The LOS is a test of voluntary movement of ones' center of gravity to the edges of their base of support. Utilizing feedback from a computer screen, participants are asked to transfer their center of gravity (COG) toward 8 targets spaced at 45-degree angular intervals around the body's COG in the 4 cardinal directions and their diagonals. The participant starts in the forward (12:00) position and then moves sequentially in a clockwise manner, covering each of the 8 directions. During the leaning motion, they are to maintain a straight posture and keep their feet planted on the floor, only moving at their ankles, like an inverted pendulum. An average of all directions is calculated for each of the following measures: a) Reaction Time: time

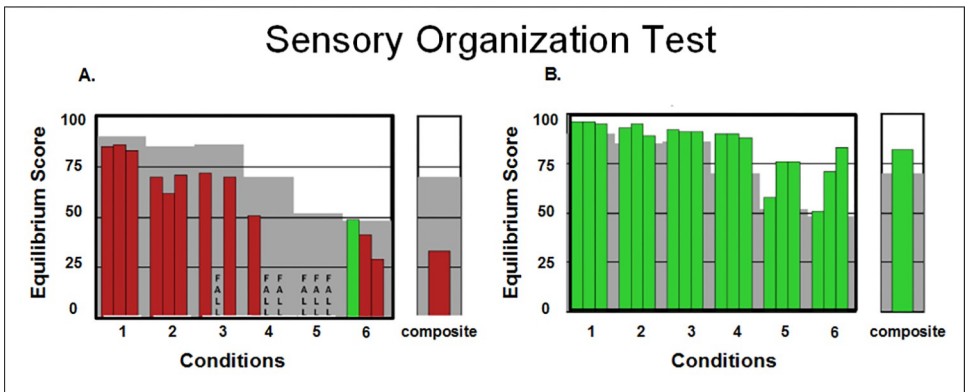

**Fig 1. Representative SOT results.** This figure shows NeuroCom reports of two participants with TBI (A. abnormal SOT performance, B. normal SOT performance). Red bars represent abnormal scores and green bars represent normal scores. Gray shading represents abnormal area that is two standard deviations from normative mean values. The individual in A, as an example, had difficulty completing the SOT test as they scored abnormal on all trials of all conditions except for trial 1 of condition 6. This individual lost balance a few times, as indicated by the word "FALL" replacing the bar on conditions 3 (trial 2), 4 (trials 2 and 3) and 5 (all trials). The individual in B had no deficits as they scored normal on every trial of every condition.

(seconds) between the start of each trial and participant's first movement towards the target, b) Movement Velocity (degrees of ankle flexion per second) from initial stance to final leaning position, c) Endpoint Excursion: greatest center of gravity displacement upon initial lean towards target (percentage of maximal theoretical limits), d) Maximum Excursion: greatest center of pressure displacement reached throughout the entire 8 second duration of the trial, and e) Directional Control: percentage of movement in the most direct path of center of pressure displacement towards the target. Fig 2 shows an illustration of the test and representative data for two participants with distinct performances.

SOT and LOS metrics were compared to the NeuroCom age-referenced normative dataset provided by the NeuroCom manufacturer and were classified as clinically abnormal if greater than two standard deviations from the mean. All subjects in the normative data set were reported to have no current or past diagnosis or injury affecting balance; be taking no medications affecting the central nervous system or known to affect balance or coordination; and have no symptoms of dizziness or light-headedness, no symptoms suggestive of vestibular or other neurologic disorders, no psychological disorders including depression, no history of two or more unexplained falls within the past 6 months, and normal vision with or without glasses. The SOT normative data included a group of 112 individuals between the ages of 20–59 years old (gender not provided by manufacturer). The means and standard deviations of their SOT parameters were equilibrium score 1: 93.9 (2.35), Equilibrium score 2: 92.0 (4.2), Equilibrium score 3: 91.5 (3.3), Equilibrium score 4: 82.5 (7.5), Equilibrium score 5: 69.2 (10.4), Equilibrium score 6: 67.2 (11.6), and Composite score 79.8 (5.6). The LOS normative data included a group of 47 healthy individuals between the ages of 40–59 years old (20 males and 27 females). The means and standard deviations of their LOS parameters are as follows: Reaction Time 0.7 (0.17), Movement Velocity 5.0 (1.5), Endpoint Excursion 84.9 (8.3), Maximum Excursion 98.0 (5.9) and Directional Control 75.2 (6.0).

## Image acquisition and processing

Structural MR and diffusion weighted images (DWIs) were acquired on a 3 tesla MR scanner (Siemens Biograph mMR) with a 16-channel head coil. DWIs were acquired using an echo

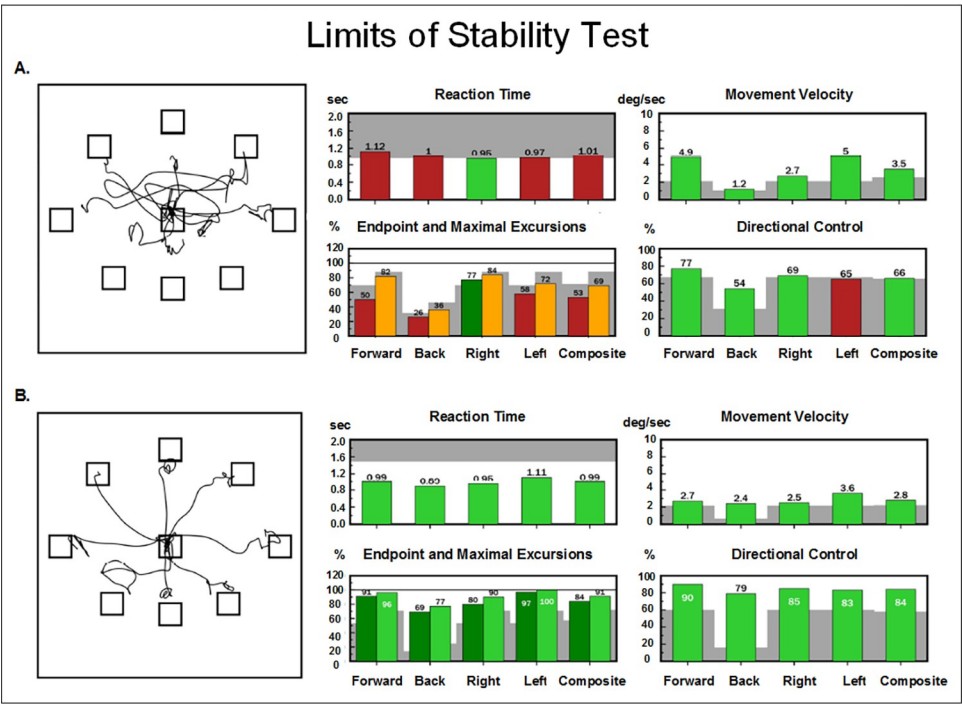

**Fig 2. Representative LOS results.** This figure shows NeuroCom reports of two participants with TBI (A. abnormal LOS performance, B. normal LOS performance). The left plots show the participant's center of gravity tracings for each trial of the LOS test. The participant is to start in the center box and lean toward each target in a clockwise manner. Ideally, the tracing should look like a star, with 8 lines radiating from the center to each target. The bar plots on the right show each participant's scores on the forward, back, right, left directions and composite scores (all directions) for each LOS variable measured (e.g. reaction time, movement velocity, endpoint and maximal excursions and directional control). Red and yellow bars represent abnormal scores and green bars represent normal scores. Gray shading represents abnormal performance that is two standard deviations from normative mean values. The individual in A had difficulty completing the LOS test as they scored abnormal on several measures, while the individual in B had no deficits scoring normal on all measures.

planar imaging sequence with parameters TR = 17000 mms, TE = 98 mms, flip angle = 90 degrees, voxel size = 2 x 2 x 2 mm, FOV = 230x230mm, GRAPPA = 2, matrix size = 128 x128, bandwidth = 1775 Hz/pixel, and slices = 75. The acquisition included a total of 80 DWIs with 10 images at b = 0 s/mm$^2$, 10 images with non-collinear directional gradients uniformly distributed on a sphere at b = 300 s/mm$^2$, and 60 images with non-collinear directional gradients uniformly distributed on a sphere at b = 1100 s/mm$^2$.

DWIs were processed using the TORTOISE software for tensor estimation [21]. Briefly, images were pre-processed for motion correction and eddy current correction, with adjustments to the gradient table performed based on participant position. Distortions due to echo planar imaging susceptibility artefacts were corrected by first performing brain extraction on an anatomic T$_2$-weighted SPACE acquisition (TR = 3200 mms, TE = 280 mms, flip angle = 120 degrees, spatial resolution = 0.98 x 0.98 x 1 mm, resampled to 0.49 x 0.49 x 1 mm). Next, a rigid registration was performed that aligned the T$_2$-weighted image to the b = 0 image using the ANTS software package [22]. Finally, a deformable registration was performed within TORTOISE from the b = 0 image to the T$_2$-weighted image, and the resulting transformation was applied to each gradient direction. After distortion correction, non-linear least-squares tensor estimation was performed followed by computation of fractional anisotropy (FA). We selected focusing on FA and not including additional DTI metrics in our main

analysis because FA is a sensitive marker of WM integrity. Furthermore, including FA also allow us direct comparison with previous DTI and balance study in children [18]. Although we also obtained other DTI metrics such as axial diffusivity, radial diffusivity and mean diffusivity for the larger natural history study, including that data in this analysis would have contributed to large number of corrections for multiple comparisons, which would have washed out any potential findings. Nevertheless, that data is presented as supplementary information in Appendix S2 (Tables 6–11 in S2 File). Measurements were averaged within all regions of interest (ROI), defined automatically using the DOTS tract segmentation algorithm [23]. The DOTS algorithm applies a Markov random field approach to segmenting 39 total white matter tracts based on the structural connectivity of each subject.

**Regions of interest (ROIs).** A recent review of the literature has shown imaging-balance relationships in several brain regions [24]. In line with our first objective to confirm associations between balance measures and DTI as previously reported, we first selected the same ROIs as those investigated in studies involving the SOT in children [18], including the corticospinal tract, inferior cerebellar peduncle, medial lemniscus, middle cerebellar peduncle and posterior thalamic radiation. Given that our second aim was exploratory in nature, and the fact that the LOS is more physically and cognitively demanding than the SOT, we selected the following additional ROIs: superior cerebellar peduncles, cingulum, corpus callosum, corticopontine tract, fronto-occipital fasciculi, inferior and superior longitudinal fasciculi, optic radiations, optic tracts, and thalamic radiations. Bilateral tracts were averaged for the purposes of this analysis. The rationale for selecting these additional ROIs is that they are involved in the demands of the LOS postural task, which include dynamic voluntary control and perception of body sway, visuospatial and auditory attention, and motor learning.

## Statistical analysis

All analyses were performed in IBM SPSS Statistics for Windows (v.28, IBM Corp., Armonk, NY, USA). Descriptive statistics were performed to report participants' characteristics. Data were checked for normality. A one-tailed Sign Test was used to compare balance scores on the SOT and LOS to the normative data provided by the NeuroCom manufacturer. The level of significance was set at alpha <0.05. Partial rank correlations were calculated, controlling for age, to determine relationships between SOT or LOS balance scores and DTI FA scores. We used averaged FA scores for bilateral tracts. Relationships between the SOT and LOS balance scores were also explored with partial rank correlations controlling for age. The Benjamini-Hochberg method was used to control for false discovery rate for multiple testing for all correlation analyses [25]. Details on the Benjamini-Hochberg analysis are presented as Appendix S1. Depending on the number of tests, correlation coefficients with adjusted $p$ value $\leq 0.013$ and $p$ value $\leq 0.023$ were regarded as significant. The computation of adjusted p values is provided in Appendix S1. A complementary correlation analysis without correction for multiple comparison was performed between SOT and LOS balance measures and DTI measures of axial diffusivity, radial diffusivity and mean diffusivity.

## Results

## Participant characteristics

Of the 172 participants enrolled in the natural history study between August 2011 and January 2019, 56 met criteria for this cross-sectional analysis. Participants (median age 43 ± 17 years old; 39 male/17 female; 71% white race). The majority of the cohort (51%) was tested between 30 and 180 days post-injury, followed by 32% tested at the 1-year time point, 7% at 2 years, 4% at 3 years, 5% at 4 years, and 1% at 5 years. Most of the sample were classified as mild TBI

**Table 1. Demographic characteristics (N = 56).**

| | Total n (% or SD) |
|---|---|
| Sex (M) | 39 (69.6%) |
| Median/Mean age (years) | 43.0/45.1 (17) |
| Race | |
| • Asian | 1 (1.8%) |
| • Black or African American | 10 (17.9%) |
| • Multiple races | 5 (8.9%) |
| • White | 40 (71.4%) |
| Ethnicity | |
| • Latino or Hispanic | 7 (12.5%) |
| • Not Latino or Hispanic | 49 (87.5%) |
| TBI Severity ** | |
| • Mild | 29 (51.8%) |
| • Moderate | 19 (33.9%) |
| • Severe | 7 (12.5%) |
| • Missing | 1 (1.8%) |
| Mechanism of injury | |
| • Acceleration/deceleration | 16 (28.6%) |
| • Blast | 3 (5.4%) |
| • Combined | 3 (5.4%) |
| • Direct impact–blow to head | 11 (19.6%) |
| • Direct impact–head against object | 6 (10.7%) |
| • Fall (ground floor) | 8 (14.3%) |
| • Fall (height >1 meter) | 9 (16.0%) |

Note: ** TBI severity is missing for one participant, combined = Acceleration/deceleration + blow to head (n = 1), Direct impact–head against object + fall (ground floor) (n = 1), Direct impact–blow to head + fall (height >1 meter) (n = 1); MVA = motor vehicle accident; Other = non-intentional injury (n = 1) and violence/assault cases (n = 3).

(53%), followed by moderate (34%) and severe (13%); classification was missing for one participant. The most common mechanism of injury was acceleration/deceleration (29%), followed by direct impact–blow to head (20%). See Table 1 for detailed demographic characteristics of the sample.

## Balance abnormalities

Table 2 shows results of the SOT and LOS tests. Group analysis comparing our cohort to the NeuroCom normative values revealed significantly lower SOT equilibrium scores for patients on conditions 2, 4 and 5. Individual data comparisons showed 23% of participants scored abnormal (e.g. at least one abnormal trial) on Equilibrium score 1, 32% on Equilibrium score 2, 34% on Equilibrium score 3, 29% on Equilibrium score 4, 42% on Equilibrium score 5, 38% on Equilibrium score 6, and 36% had abnormal Composite scores. Finally, 25% had normal scores on all variables. Representative data is shown in Figs 1 and 2.

Group comparisons for the LOS revealed significant deficits for participants on all measures. Individual data comparisons showed 41% scored abnormal (e.g. abnormal composite scores) on Reaction Time, 27% on Movement Velocity, 38% on Endpoint Excursion, 51% on Maximal Excursion and 2% demonstrated abnormal Directional Control. Finally, 29% (n = 16) achieved normal scores on all variables.

Further examination of the data showed that 16% of the cohort tested normal on both the SOT and LOS tests. That subgroup of individuals was heterogeneous in terms of age, sex and TBI chronicity (e.g. time of post-injury testing varied between 30 days and 2 years). TBI classification was mild and moderate; there were no severe TBIs in the group.

**Table 2. SOT and LOS balance assessment results in comparison to NeuroCom normative database.**

| Balance Measure | Patients (Mean ± SD) | Controls (Mean ± SD) | Significance |
|---|---|---|---|
| SOT | | | |
| Equilibrium Score 1 | 93.3 ± 3.1 | 93.7 ± 0.8 | 0.248 |
| Equilibrium Score 2 | 87.7 ± 7.9 | 91.6 ± 1.2 | **0.005** |
| Equilibrium Score 3 | 88.1 ± 7.6 | 90.9 ± 1.1 | 0.170 |
| Equilibrium Score 4 | 81.6 ± 13.3 | 82.9 ± 1.5 | **0.027** |
| Equilibrium Score 5 | 54.1 ± 13.2 | 68.2 ± 2.0 | **0.049** |
| Equilibrium Score 6 | 60.6 ± 22.5 | 66.2 ± 2.8 | 0.292 |
| Composite Score | 73.8 ± 12.6 | 79.1 ± 1.5 | 0.392 |
| LOS | | | |
| Reaction Time | 0.9 ± 0.2 | 0.7 ± 0.1 | **<0.001** |
| Movement Velocity | 3.7 ± 1.5 | 5.1 ± 0.7 | **<0.001** |
| Endpoint Excursion | 69.9 ± 15.5 | 82.1 ± 5.7 | **<0.001** |
| Maximal Excursion | 81.7 ± 13.5 | 94.8 ± 4.0 | **<0.001** |
| Directional Control | 77.9 ± 6.5 | 73.0 ± 1.6 | **<0.001** |

Note: Controls=NeuroCom normative database. Group size was 56 for all variables except Equilibrium Score 4-6 and Composite scores (n=55 because one participant felt nauseated and had to stop testing). Bold values indicate significant difference at p<0.05. Abbreviations: SOT=Sensory Organization Test, LOS=Limits of Stability Test.

## SOT and DTI FA associations

SOT and DTI FA associations are presented in Table 3. After correcting for multiple comparisons with the Benjamini-Hochberg method, several WM tracts were significantly associated with SOT Equilibrium scores 4–6 and with Composite scores. The inferior cerebellar peduncle correlated with Equilibrium scores 4, 5, 6, and Composite score. The middle cerebellar peduncle correlated significantly with Equilibrium scores 4, 6 and Composite score. The medial lemniscus correlated significantly with Composite score. The posterior thalamic radiation correlated significantly with Equilibrium score 6 and Composite score. Direction of association was positive for all significant correlations; lower values for SOT Equilibrium and

**Table 3. Correlations between SOT and FA, controlled for age (n = 52).**

| ROI | Statistics | EqC1 | EqC2 | EqC3 | EqC4 | EqC5 | EqC6 | Comp |
|---|---|---|---|---|---|---|---|---|
| Corticospinal tract | r | 0.153 | -0.101 | 0.062 | 0.249 | 0.172 | 0.217 | 0.219 |
| | p-value | 0.268 | 0.469 | 0.654 | 0.069 | 0.213 | 0.115 | 0.112 |
| Inferior cerebellar peduncle | r | 0.253 | -0.003 | 0.226 | **0.364** | **0.380** | **0.357** | **0.402** |
| | p-value | 0.065 | 0.980 | 0.100 | 0.007 | 0.005 | 0.008 | 0.003 |
| Middle cerebellar peduncle | r | 0.085 | 0.068 | 0.214 | **0.337** | 0.303 | **0.450** | **0.382** |
| | p-value | 0.539 | 0.626 | 0.120 | 0.013 | 0.026 | 0.001 | 0.004 |
| Medial lemniscus | r | 0.146 | 0.016 | 0.252 | 0.255 | 0.296 | 0.278 | **0.317** |
| | p-value | 0.291 | 0.910 | 0.066 | 0.063 | 0.030 | 0.042 | 0.019 |
| Posterior thalamic radiation | r | 0.189 | 0.164 | 0.282 | 0.296 | 0.308 | **0.344** | **0.352** |
| | p-value | 0.172 | 0.237 | 0.039 | 0.029 | 0.023 | 0.011 | 0.009 |

Note: Bold values indicate significant correlations at p≤0.013 following Benjamini-Hochberg method. Abbreviations: EqC1=Equilibrium Score condition 1, EqC2= Equilibrium Score condition 2, EqC3= Equilibrium Score condition 3, EqC4= Equilibrium Score condition 4, EqC5= Equilibrium Score condition 5, EqC6= Equilibrium Score condition 6, Comp= composite score.

Composite scores were associated with lower FA values. Correlations between SOT and other DTI parameters are presented in Appendix S2 as complementary information. As can be seen, most p-values would not have survived corrections for multiple comparison.

## LOS and DTI FA associations

After correcting for multiple comparisons with the Benjamini-Hochberg method, numerous WM tracts were significantly associated with LOS parameters. Specific correlation and significance values are presented in Table 4. FA of the corpus callosum stood out as the region with the strongest and largest number of associations, as its posterior fibers correlated with all LOS

**Table 4. Correlations between LOS and FA, controlled for age (n = 53).**

| ROI | Statistics | RT | MVL | EPE | MXE | DCL |
|---|---|---|---|---|---|---|
| Anterior thalamic radiation | r | -0.266 | 0.168 | **0.374** | **0.318** | **0.354** |
| | p-value | 0.050 | 0.219 | 0.005 | 0.018 | 0.008 |
| Corpus callosum-anterior | r | -0.287 | **0.326** | **0.440** | **0.423** | **0.398** |
| | p-value | 0.034 | 0.015 | 0.001 | 0.001 | 0.003 |
| Corpus callosum-posterior | r | **-0.441** | **0.379** | **0.414** | **0.397** | 0.269 |
| | p-value | 0.001 | 0.004 | 0.002 | 0.003 | 0.047 |
| Corpus callosum-superior | r | -0.283 | 0.272 | **0.332** | **0.371** | 0.234 |
| | p-value | 0.036 | 0.044 | 0.013 | 0.005 | 0.086 |
| Cingulum | r | **-0.335** | 0.280 | **0.370** | **0.313** | 0.303 |
| | p-value | 0.012 | 0.038 | 0.005 | 0.020 | 0.025 |
| Corticopontine tract | r | **-0.344** | 0.126 | 0.266 | 0.146 | 0.265 |
| | p-value | 0.010 | 0.361 | 0.049 | 0.289 | 0.051 |
| Corticospinal tract | r | **-0.387** | 0.259 | **0.361** | 0.226 | 0.164 |
| | p-value | 0.004 | 0.056 | 0.007 | 0.097 | 0.231 |
| Inferior cerebellar peduncle | r | **-0.321** | **0.376** | **0.373** | **0.350** | **0.387** |
| | p-value | 0.017 | 0.005 | 0.005 | 0.009 | 0.004 |
| Inferior fronto-occipital fasciculus | r | -0.291 | 0.248 | **0.350** | **0.366** | 0.259 |
| | p-value | 0.031 | 0.067 | 0.009 | 0.006 | 0.056 |
| Inferior longitudinal fasciculus | r | **-0.365** | **0.308** | **0.380** | **0.379** | 0.263 |
| | p-value | 0.006 | 0.022 | 0.004 | 0.004 | 0.053 |
| Middle cerebellar peduncle | r | -0.283 | 0.162 | 0.182 | 0.113 | 0.331 |
| | p-value | 0.037 | 0.238 | 0.183 | 0.411 | 0.014 |
| Medial lemniscus | r | **-0.349** | **0.409** | 0.265 | 0.258 | 0.174 |
| | p-value | 0.009 | 0.002 | 0.051 | 0.058 | 0.203 |
| Optic radiation | r | -0.254 | 0.196 | 0.290 | 0.275 | 0.256 |
| | p-value | 0.061 | 0.152 | 0.032 | 0.042 | 0.059 |
| Optic tracts | r | **-0.372** | 0.264 | **0.358** | 0.272 | **0.325** |
| | p-value | 0.005 | 0.052 | 0.007 | 0.045 | 0.016 |
| Posterior thalamic radiation | r | -0.186 | 0.140 | 0.219 | 0.218 | **0.392** |
| | p-value | 0.174 | 0.309 | 0.108 | 0.110 | 0.003 |
| Superior fronto-occipital fasciculus | r | -0.120 | -0.022 | 0.080 | 0.087 | 0.117 |
| | p-value | 0.382 | 0.876 | 0.561 | 0.529 | 0.395 |
| Superior longitudinal fasciculus | r | **-0.460** | 0.276 | 0.292 | 0.241 | **0.326** |
| | p-value | 0.000 | 0.041 | 0.031 | 0.077 | 0.015 |

Note: Bold values indicate significant correlations at p≤0.023 following Benjamini-Hochberg method. Abbreviations: R-right, L-left, RT-reaction time, MVL-movement velocity, EPE-endpoint excursion, MXE-maximal excursion, DCL-directional control

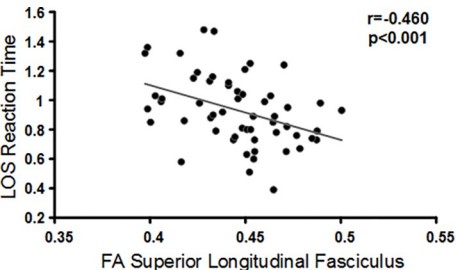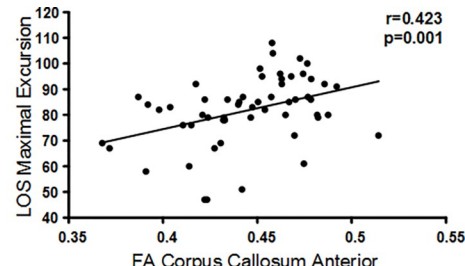

**Fig 3. Representative correlations.** These scatterplots are representative of the strength and direction of associations between balance and DTI measures. Left scatterplot: Negative correlation between LOS Reaction Time and FA of the superior longitudinal fasciculus. Right scatterplot: Positive correlation between LOS Directional Control and FA of the corpus callosum.

variables except for LOS Directional Control. Its anterior fibers correlated with Movement Velocity, Endpoint Excursion, Maximal Excursion, and Directional Control. Its superior fibers correlated with Endpoint Excursion and Maximal Excursion. In addition to the corpus callosum, two other areas also showed a large number of associations: inferior cerebellar peduncle, which correlated with all LOS variables, and the inferior longitudinal fasciculus, which correlated with all LOS variables except for LOS Directional Control. Correlations between LOS and other DTI parameters are presented in the Appendix S2 as complementary information. As can be seen, most p values would not have survived corrections for multiple comparisons.

Other areas with significant associations were: 1) the anterior thalamic radiation, which correlated with Endpoint Excursion, Maximal Excursion, and Directional Control; 2) cingulum, which correlated with Reaction Time, Endpoint Excursion and Maximal Excursion; 3) corticopontine tract, which correlated with Reaction Time; 4) corticospinal tract, which correlated with Reaction Time and Endpoint Excursion; 5) inferior fronto-occipital fasciculus, which correlated with Endpoint and Maximal Excursion; 6) medial lemniscus, which correlated with Reaction Time and Movement Velocity; 7) optic tracts, which correlated with Reaction Time and Endpoint Excursion; 8) posterior thalamic radiation, which correlated with Directional Control; and 9) superior longitudinal fasciculus, which correlated with Reaction Time and Directional Control. Direction of association was positive for all significant correlations except for LOS Reaction Time/FA correlations, for which lower FA values were correlated with higher RT values. RT differs from other measures in that a higher value (longer reaction time) indicates poorer performance.

Fig 3 is representative of an inverse and a direct correlation between balance and WM integrity. The left scatterplot shows a negative association between LOS Reaction Time and FA of the superior longitudinal fasciculus tract (r = -0.460, p = 0.001), such that as Reaction Time values decrease (better balance), FA values increase (better WM integrity). The right scatterplot shows a positive association between LOS Maximum Excursion and FA of the corpus callosum (r = 0.423, p = 0.001), such that as Maximum Excursion increases (better balance), FA values increase.

## SOT and LOS associations

Table 5 shows associations between SOT and LOS measurements after correcting for multiple testing with the Benjamini-Hochberg method. It can be observed that the SOT composite score and the more challenging SOT conditions (Equilibrium score conditions 4–6) were significantly associated with LOS measures of Reaction Time, Movement Velocity, Endpoint Excursion and Maximal Excursion. Additionally, LOS Endpoint Excursion was associated with Equilibrium score 3. There were no associations between SOT and Directional Control.

**Table 5. Correlations between SOT and LOS, controlled for age (n = 52).**

| Limits of Stability Test | Statistics | Sensory Organization Test | | | | | | |
| --- | --- | --- | --- | --- | --- | --- | --- | --- |
| | | EqC1 | EqC2 | EqC3 | EqC4 | EqC5 | EqC6 | Comp |
| RT | r | -0.172 | 0.063 | -0.219 | **-0.451** | **-0.406** | **-0.507** | **-0.464** |
| | p-value | 0.213 | 0.653 | 0.112 | 0.001 | 0.002 | 0.000 | 0.000 |
| MVL | r | 0.188 | -0.095 | 0.288 | **0.318** | **0.336** | **0.363** | **0.351** |
| | p-value | 0.174 | 0.496 | 0.035 | 0.019 | 0.013 | 0.007 | 0.009 |
| EPE | r | 0.265 | 0.170 | **0.329** | **0.333** | **0.451** | **0.442** | **0.411** |
| | p-value | 0.053 | 0.218 | 0.015 | 0.014 | 0.001 | 0.001 | 0.002 |
| MXE | r | 0.165 | 0.126 | 0.299 | 0.165 | **0.382** | **0.397** | **0.331** |
| | p-value | 0.234 | 0.365 | 0.028 | 0.233 | 0.004 | 0.003 | 0.014 |
| DCL | r | 0.220 | 0.118 | 0.254 | 0.173 | 0.234 | 0.293 | 0.268 |
| | p-value | 0.110 | 0.396 | 0.064 | 0.210 | 0.089 | 0.031 | 0.050 |

Note: Bold values indicate significant correlations at p≤0.023 following Benjamini-Hochberg method. Abbreviations: RT-Reaction Time, MVL-Movement Velocity, EPE-Endpoint Excursion, MXE-Maximal Excursion, DCL-Directional Control, EqC1=Equilibrium score condition 1, EqC2= Equilibrium score condition 2, EqC3= Equilibrium score condition 3, EqC4=Equilibrium score condition 4, EqC5= Equilibrium score condition 5, EqC6= Equilibrium score condition 6, Comp= Composite score.

## Discussion

This is the first study establishing relationships between postural control and white matter integrity in adults post-TBI. Our results clearly show that reduced WM integrity in multiple brain tracts is correlated with poorer balance performance post-TBI. Related to our first objective, we replicated many of the associations found in children involving the SOT and FA measures of the medial lemniscus and the middle cerebellar peduncle. Our second aim was to explore performance on the LOS and associations with WM tract integrity post-TBI. Several ROIs were identified with significant associations between FA and LOS measures. Among them the corpus callosum stood out as the region with the largest number of associations, as FA of the different portions of this region correlated with all 5 LOS measures. Additionally, we compared ROIs associated with SOT versus LOS and found these balance tests shared a few overlapping tracts, namely the inferior cerebellar peduncle, medial lemniscus and posterior thalamic radiation.

The cohort investigated in this study was heterogeneous in terms of age, time post-injury, classification of TBI severity and mechanism of injury. They also presented with considerable variations in balance scores on both the SOT and LOS tests. As a group, their scores were significantly worse than controls; however, examination of individual data revealed a wide range of scores with approximately 25% of participants performing normally on each test. Our results agree with the literature showing that balance deficits affect a large proportion, but not all, of those who have suffered a TBI [11,26]. As much as a homogeneous group is desirable methodologically, this wide range of balance scores is advantageous for correlation studies such as ours because it allows for better spread of the datapoints and represents the observed variability that exists in balance performance among the larger TBI population.

Our SOT findings partially agree with the observations of Caeyenbergh et al. [18]. Two ROIs identified in their work were also significantly associated with the SOT in our study: the medial lemniscus and the middle cerebellar peduncle. However, while Caeyenberghs et al. [18] reported associations with baseline SOT conditions (1 and 3), we are reporting associations with more challenging SOT conditions (4, 5 and composite). Such inconsistencies are not surprising given the differences between datasets in terms of sample size and age, as we know

children's SOT scores change as they age and only reach adult-like levels in their late teen years [35]. The upper-level SOT conditions found to be significant in our study are more indicative of the use and integration of specific sensory inputs for postural control. Our findings make sense in the context of the existing literature as both the medial lemniscus and the middle cerebellar peduncle are known to be involved in the multisensory integration process that is needed for the control of standing balance [27]. Moreover, as part of the brainstem and the cerebellum, these two tracts are the most frequently implicated in associations with standing balance in the literature as recently reported by Surgent et al. [24]. The medial lemniscus is one of the main somatosensory pathways conveying information about proprioception through the brainstem and terminating in the thalamus [28]. Degeneration in the medial lemniscus correlates with measures of disease severity and symptoms of parkinsonian syndromes [29]. Also, studies in children with hemiplegia have shown that deterioration of the medial lemniscus without damage to the corticospinal tract results in balance impairments [30,31]. The middle cerebellar peduncle is an afferent tract carrying somatosensory information to the cerebellum [27]. Degeneration of the middle cerebellar peduncle has been linked to multiple system atrophy, a parkinsonian syndrome which greatly affects postural instability [32], and to cases of severe postural ataxia post-TBI [33]. Additionally, integrity of the middle cerebellar peduncle has been reported to predict balance training improvement in children post-TBI who practice computer-based dynamic balance tasks similar to the LOS [34]. In conjunction with these studies from other groups, it is clear from our findings that the medial lemniscus and middle cerebellar peduncle play key functional roles in standing balance in both children and adults.

A possible explanation for the discrepancy between our results and those by Caeyenberghs et al. [18] may be related to differences in age of the participants. As previously mentioned, maturity of postural control is not achieved until adulthood. SOT Equilibrium scores vary with age and gender for all conditions of the test. From age 5 to 16 years, SOT performance changes to a clinically significant degree every 2–3 years and is significantly different from adults at every age group, including adolescents (16 years old) [35]. The ages of the children studied by Caeyenberghs et al. ranged between 8 and 20 years old [18]. Not surprisingly, our participants, who were all adults, performed differently on the SOT and reflected impairments in higher-order balance conditions as might be expected for older individuals with more mature postural control. In regards to the contrasting findings of WM tracts affected in our sample versus those found to be affected in the pediatric sample [18], it should be noted that only 12 patients with TBI were included in the pediatric study, creating opportunity for significant variation in injury patterns between patients and opportunities for a relatively small number of patients in the sample to markedly influence the overall results. Additionally, the pediatric patients were all classified as moderate-severe TBI, whereas our sample consisted of a mix of mild to severe TBI with mild TBI being most common. Accordingly, different WM tracts may have been altered by TBI in our cohort due to the wider range of injury severity, and as a result, differing associations were likely observed between reduced WM integrity and poorer performance on the SOT.

Our study also identified areas associated with the SOT that were not observed by Caeyenberghs et al. [18] including the inferior cerebellar peduncle and the posterior thalamic radiation. Their study mentions, however, that some of these ROIs showed strong associations with SOT performance but did not survive multiple comparison corrections. In our study these ROIs were not only associated with the SOT but also the LOS. These findings make sense given the SOT and the LOS are themselves associated, and given their role in both postural control during quiet stance (SOT) and during dynamic postural tasks (LOS). The associations we found between the SOT and LOS were expected given that the formula to calculate one's

SOT Equilibrium Scores takes into account their LOS Maximal Excursion Scores [36]. The associations we observed between the most challenging SOT conditions (Equilibrium scores 4–6 and also Composite scores) and LOS variables make sense as it is during the most challenging SOT conditions that patients sway at their maximum amplitudes in terms of excursion and velocity.

The inferior cerebellar peduncle carries somatosensory information to the cerebellum and is critical for postural control during standing with or without visual input, as evidenced by investigations of postural sway and DTI in multiple sclerosis [37]. Finally, the fundamental role of the posterior thalamic radiation on the control of upright body posture is well illustrated by reports in patients with "pusher syndrome" (impaired subjective vertical sensation) [38] as well as parkinsonian syndromes, which all share disturbances of balance and gait as a hallmark [39,40].

As the main connection between primary motor cortex, supplementary motor areas and the spinal cord, the corticospinal tract is the major efferent output involved in the maintenance of standing posture and initiation of intentional movement [41]. In particular, our finding of correlations between the corticospinal tract and LOS Reaction Time and Endpoint Excursion are supported by studies in video game players performing a visual attention task [42], as well as studies in older individuals with poor postural reactions [43]. Surprisingly, though, we did not find a significant correlation between the corticospinal tract and any of the SOT components.

As mentioned previously, the LOS test has a heavy visual feedback component. Participants rely on visual feedback to move their center of gravity toward the target. Not surprisingly, the optic tracts were correlated with LOS Reaction Time, Endpoint Excursion and Directional Control, and the posterior thalamic radiation correlated with LOS directional control. The optic tracts form the anterior visual pathway, as they constitute the continuation of the optic nerves after the partial decussation at the optic chiasm. Clear evidence of the important role of vision in postural control comes from research in cerebral palsy, which often causes visual dysfunction of the anterior or posterior portion of the visual system and greatly affects postural control [44]. Moreover, integrity of the same WM tracts was associated with visuomotor tracking performance scores in children performing a manual pursuit test of eye-hand coordination [45]. Although the task in their study did not involve standing balance, it required some overlapping constructs with the LOS test, such as attention, visual spatial skills as well as proprioception. Their study also showed correlations with the corticospinal tract, which is consistent with our findings.

The superior longitudinal fasciculus is also related to visual function, as it plays a role in the regulation of visual attention and higher visual processing [46–48]. This tract provides the main source of information transfer between frontal and parietal cortices and it relates to balance as it provides the exchange of information about one's perception of the body in space for planning, initiation and updating of goal-directed movement [49]. White matter integrity of the superior longitudinal fasciculus correlates with manual dexterity and finger tapping tasks in stroke [50]. In healthy individuals this tract participates in planning of reaching and grasping movements [51]. It is also involved in information processing speed, as reduced FA of this tract correlates with slower processing speed in older adults [52]. Our findings of associations between FA of the superior longitudinal fasciculus and LOS Reaction Time may be specifically due to the speed component involved in this LOS measure, but may also reflect the broad contribution of this pathway in goal-directed motor functioning, as is likely the case for its role in the directional control measure.

The corpus callosum was correlated with every LOS measure. There is speculation that this area may be particularly vulnerable to diffuse axonal injury, as it is the most common location

of abnormal FA in TBI [53]. Research on balance and DTI in TBI, though, had not explored the corpus callosum prior to this study. As such, we are limited to comparing our findings to non-TBI populations. Our results are in line with studies in the field of aging reporting associations between FA of the corpus callosum and poor balance on the SOT [54] and the Tinetti scale [55] in older adults with gait and mobility problems. Additionally, there is evidence that the corpus callosum is involved in visuospatial attention [56,57]–a crucial skill in the control of sway direction toward each target and excursion on the LOS test.

Finally, the anterior thalamic radiation, which was correlated with LOS endpoint excursion, maximum excursion and directional control, has been shown to be involved in the planning of complex behaviours [58–60]. As successful excursion and directional control certainly depend on adequate planning of complex motor behaviour, our results seem to indicate the anterior thalamic radiation contributes in these two different, yet complementary, aspects of postural control on the LOS test.

This research has several limitations, which include a relatively small and heterogeneous sample, the lack of a control group and the large number of multiple comparisons. Although the size of our cohort is larger than that of any prior study of this kind, additional studies are needed to confirm our findings in larger cohorts. A larger sample size would have accommodated for more robust statistical analyses such as regression models controlling for the effects of TBI severity and time from injury. It would also allow for the use of broader and unbiased whole brain white matter techniques, such as tract-based spatial statistics. This technique would be especially pertinent to LOS associations since the LOS balance test also involves cognitive circuits that could not be fully represented with our analysis. Adding cognitive impairment measures as covariates to more robust analyses would also be worth exploring. We also acknowledge the lack of a control group in this study. We used manufacturer-provided normative data to understand the extent of the balance abnormalities but did not have imaging data from healthy individuals for comparative analyses. The main purpose of this study, however, was to focus on TBI and investigate whether WM damage was associated with impairments in balance, and for that correlation analysis a control group was not imperative.

Related to the use of DTI and its analysis, another limitation is that the estimation of diffusion properties does not model crossing fibers in white matter. Such fibers may result in biased measurements within ROIs, although it is not clear how much this effect will differ across subjects. Furthermore, we performed a large number of correlations, but due to the exploratory nature of this study it was necessary to select this list of ROIs. We did, however, correct for multiple comparisons. Had we limited ROIs *a priori*, we would also be limiting the possibility of identifying important pathways related to postural control especially those involved in LOS. As previous research has shown, a large number of structures across the brain are associated with balance [24]. To reduce the number of comparisons, we focused only on FA, which has been the most widely used DTI metric for TBI studies [61,62]. Mean diffusivity, radial diffusivity, and axial diffusivity are additional common metrics that have been shown to have value in TBI studies, so we reported those results complementary material without correcting them for multiple comparisons.

## Conclusions

In conclusion, our study contributes to the understanding of balance alterations post-TBI as it utilizes two distinct yet complementary balance tests to investigate balance performance in association with imaging. We have confirmed, in part, findings in children showing poorer balance performance on the SOT was correlated with reduced WM integrity. We have also identified key structures that are likely to underlie postural control in both the SOT and LOS

tasks. Moreover, we found that poorer performance on all measures of the LOS was associated with impaired WM integrity, especially of tracts involved in planning and execution of complex movements, information processing speed, somatosensory integration and visual function. These findings show that, in adults with TBI, standing balance control seems to depend upon the integrity of both long-and short-range WM fiber bundles involved in a broad range of brain functions.

## Supporting information

**S1 File. Benjamini-Hochberg (B-H) procedure.** Description of the computation of B-H corrected p values for each set of correlations.
(DOCX)

**S2 File. Correlations between balance and additional DTI measures.** These are complementary tables showing correlation values between each balance measure and DTI measures that were not included in the main manuscript (Axial Diffusivity, Medial Diffusivity and Radial Diffusivity).
(DOCX)

## Acknowledgments

Thanks to Andre van der Merwe, at the Center for Neuroscience and Regenerative Medicine, for coordinating this project.

## Author Contributions

**Conceptualization:** Cris Zampieri, Pashtun Shahim, Diane Damiano, Leighton Chan.

**Data curation:** Jacob B. Leary, Pashtun Shahim, Dzung L. Pham.

**Formal analysis:** Pei-Shu Ho.

**Funding acquisition:** Leighton Chan.

**Methodology:** Cris Zampieri, Pashtun Shahim, Dzung L. Pham, Leighton Chan.

**Writing – original draft:** Cris Zampieri, Jacob B. Leary, Pashtun Shahim, Leighton Chan.

**Writing – review & editing:** Cris Zampieri, Jacob B. Leary, Pashtun Shahim, Diane Damiano, Pei-Shu Ho, Dzung L. Pham, Leighton Chan.

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
