## [Decision Letter · Decision Letter 0]

20 Apr 2023

PONE-D-23-06025Associations between brain white matter integrity and postural control in adults with traumatic brain injuryPLOS ONE

Dear Dr. Zampieri,

Thank you for submitting your manuscript to PLOS ONE. After careful consideration, we feel that it has merit but does not fully meet PLOS ONE’s publication criteria as it currently stands. Therefore, we invite you to submit a revised version of the manuscript that addresses the points raised during the review process.

Please submit your revised manuscript by June 07 2023 11:59PM. If you will need more time than this to complete your revisions, please reply to this message or contact the journal office at plosone@plos.org. Please include the following items when submitting your revised manuscript:A rebuttal letter that responds to each point raised by the academic editor and reviewer(s). You should upload this letter as a separate file labeled 'Response to Reviewers'.A marked-up copy of your manuscript that highlights changes made to the original version. You should upload this as a separate file labeled 'Revised Manuscript with Track Changes'.An unmarked version of your revised paper without tracked changes. You should upload this as a separate file labeled 'Manuscript'.

We look forward to receiving your revised manuscript.

Kind regards,

Monika Błaszczyszyn

Academic Editor

PLOS ONE

Journal Requirements:

Reviewers' comments:

Reviewer's Responses to Questions

**Comments to the Author**

1. Is the manuscript technically sound, and do the data support the conclusions?

Reviewer #1: Partly

Reviewer #2: Yes

2. Has the statistical analysis been performed appropriately and rigorously? 

Reviewer #1: No

Reviewer #2: No

3. Have the authors made all data underlying the findings in their manuscript fully available?

Reviewer #1: No

Reviewer #2: Yes

4. Is the manuscript presented in an intelligible fashion and written in standard English?

Reviewer #1: Yes

Reviewer #2: No

5. Review Comments to the Author

Reviewer #1: I reviewed with interest the article entitled “Associations between brain white matter integrity and postural control in adults with traumatic brain injury” by Prof. Cris Zampieri. This study evaluated fifty-six participants with history of TBI using SOT and LOS and correlated it with white matter integrity measured using Fractional anisotropy.

The article is well written and answers pertinent questions. I have only few suggestions-

1. As reported by the authors, LOS has a cognitive component compared to SOT. However, an ROI based approach was used based on previous literature. Considering that there is limited literature available for hypothesis-driven approach, would it be more prudent to use an unbiased whole brain white matter technique such as tract based spatial statistics (TBSS). This is especially pertinent since LOS also involves cognitive circuits which I do not see fully represented.

2. I don’t think the statistical approach is very robust. While the authors have used partial correlation controlling for age, I think you should consider using a regression model with other important covariates (as independent variables) such as time from injury (at which measurement was performed-90 days to five years) and type of injury (mild, mod, and severe).

3. Since LOS invokes cognitive control and TBI can cause mild to severe cognitive impairment, it would be interesting to see its’ effect on LOS. If available, cognitive impairment is an important covariate to be added into regression model.

4. The DWI/DTI acquisition parameters can be better described for sake of replication by other groups. Furthermore, considering that many of the ROI selected for the study has crossing fibers, limitations related to inaccurate estimation should be mentioned. I also feel that not using other DTI metrics (RD, MD, AD) may not be in best interest of the data presented. If available, the combinations of FA, AD, MD, and RD can give a lot of information with respect to nature of WM injury (neurodegeneration, demyelination, etc.) which may be pertinent. This can be added as supplementary material if word economy is a concern.

5. The discussion could be made more informative if the injured tracts and the cortical/subcortical structures it connects is given more representation. Different components of LOS and SOT and their significance can also be described better.

Overall, this is a good study and worth reading. Thank you!

Reviewer #2: Overview:

This study analyzed the associations between the SOT/ LOS and FA of the brain WM areas in 56 participants with TBI. In addition to replicate a publicated study about the SOT and WM relationship, this study also reported significant correlations between abnormal LOS scores and impaired WM integrity in the cingulum, corpus callosum, corticopontine tracts, fronto-occipital fasciculi, longitudinal fasciculi and optic tract.

This study is of scientific interesting and may improve understandings on TBI-induced balance disturbances. However, results and discussion sections are somewhere redundant and less focusing. Major revisions maybe needed.

Major concerns:

(1) In Materials and methods: What atlas was used for ROI selection? and how to transfer the ROI-masks onto individual space? Please provide citations. Since the second objective is exploratory in nature, why not analyze all white matter ROIs in atlas to find out which specific white matter tracts are significantly correlated with LOS?

(2) In Materials and methods: It seems that the Benjamini-Hochberg multiple comparison is not used for correlation test between NeuroCom and FA, if I understood correctly. It is necessary to apply multiple comparison-adjusted p onto all correlation tests in Table 3 and 4.

(3) In Results: The first correlation test between SOT and FA is designed to replicate a previous study by Caeyenbergh et al. (ref#18). However, since the current data set has different features from Caeyenbergh’s study, in terms of age, inclusion of healthy controls, and sample size, it is not surprising to observe inconsistent findings. It would be clearer to deemphasize the discussion of the comparisons with ref#18, and focusing on the interpretations of significant relationship and neuroanatomy.

(4) In Results: In this study, the exploratory LOS performed much better than commonly accepted SOT in differentiating TBI from controls. although the authors addressed that they used the manufactory provided normative data because they don't have actual healthy controls, one may wonder whether the significant group difference could have been resulted from the substantially different feature of the normative dataset that provided by NeuroCom. Additional tests may need to help understanding whether LOS is superior to SOT in describing postural control in TBI. For example, correlation tests between SOT and LOS subcategories may help to identify the similar or disassociated measurement among these tests.

(5) In Results: Table 3 and 4 showed clearly asymmetric findings. The asymmetric findings are common in TBI but the current study has no comments on these findings. Since this manuscript is focused on symmetric postural test and white matter tracts, it is better to perform the analysis in averaged bilateral tracts, or keep the current findings and adding with a careful discussion of asymmetry.

Minor concerns:

(6) In Materials and methods: Please provide demographic details of NeuroCom age-referenced normative data set, e.g., number of controls, mean (SD) age and sex.

(7) In Materials and methods: Please be sure that No. of directions at b=300 is 10, and No. of directions at b=1100 is 60.

(8) Typo on page#4: "paediatric" should be "pediatric".

6. PLOS authors have the option to publish the peer review history of their article (what does this mean?). If published, this will include your full peer review and any attached files.

Reviewer #1: **Yes: **Albert Stezin

Reviewer #2: No

---

## [Author Response · Author response to Decision Letter 0]

8 Jun 2023

PONE-D-23-06025

Associations between white matter integrity and postural control in adults with traumatic brain injury

PLOS ONE

Dear Monika Błaszczyszyn and reviewers,

Thank you for the thorough review our manuscript. We appreciate your valuable comments and suggestions. Please find a point-by-point response to each of the reviewer comments (the line numbers reported below refer to the manuscript with track changes): 

Reviewers' comments:

Reviewer #1: I reviewed with interest the article entitled “Associations between brain white matter integrity and postural control in adults with traumatic brain injury” by Prof. Cris Zampieri. This study evaluated fifty-six participants with history of TBI using SOT and LOS and correlated it with white matter integrity measured using Fractional anisotropy.

The article is well written and answers pertinent questions. I have only few suggestions-

1. As reported by the authors, LOS has a cognitive component compared to SOT. However, an ROI based approach was used based on previous literature. Considering that there is limited literature available for hypothesis-driven approach, would it be more prudent to use an unbiased whole brain white matter technique such as tract based spatial statistics (TBSS). This is especially pertinent since LOS also involves cognitive circuits which I do not see fully represented.

Author’s response: Thank you for the suggestion. We understand that TBSS is a voxel-based analysis rather than region-based, and that TBSS would be a viable approach to exploring what regions might be affected if there was not sufficient literature to narrow our focus. However, despite the limited literature available, we believe previous research [18] gives us sufficient information to select the important regions of interest and conduct a hypothesis-driven analysis. Moreover, our primary objective was to replicate the pediatric study (18] in adults using the same methods. Changing the methods would limit us from making direct comparisons. Lastly, broadening our brain imaging technique to an unbiased whole brain analysis would result in a very large number of comparisons which our study cannot accommodate given our relatively small sample size. We are thankful for the suggestion, though, and have included it as a limitation of the study (lines 538-544). We have also edited the last paragraph of the introduction to make it clear that our first goal was to replicate prior research (lines 66-68). 

2. I don’t think the statistical approach is very robust. While the authors have used partial correlation controlling for age, I think you should consider using a regression model with other important covariates (as independent variables) such as time from injury (at which measurement was performed-90 days to five years) and type of injury (mild, mod, and severe).

Authors’ response: We appreciate your suggestion and agree that a more robust approach, such as a regression model with the covariates you pointed out, should be considered. However, this was a preliminary study with a relatively small sample size. According to previous research, reliable regression coefficients would require a large enough ratio of subjects to predictors (Pedhazur & Schmelkin, 1991; Stevens, 2002). For instance, Stevens (2002) suggested a 15:1 subject-to-predictor ratio for multiple regression analysis and Pedhazur and Schmelkin (1991) recommended a sample size 30k, where k is the number of predictors. Furthermore, since our primary objective was to replicate previous research [18] and their only covariate was age, we chose to follow the same type of analysis. The reviewer’s suggestion makes sense, though, and has been included in the last paragraph of the discussion where we talk about limitations (lines 539-540). 

Stevens, J. (2002). Applied multivariate statistics for the social sciences (4th ed.). Mahwah, NJ: Lawrence Erlbaum Associates.

Pedhazur, E. J., & Schmelkin, L. P. (1991). Measurement, design, and analysis: An integrated approach. Hillsdale, NJ: Lawrence Erlbaum Associates.

3. Since LOS invokes cognitive control and TBI can cause mild to severe cognitive impairment, it would be interesting to see its’ effect on LOS. If available, cognitive impairment is an important covariate to be added into regression model.

Author’s response: Thank you for bringing this up. Indeed, it would be interesting to see the effects of cognitive impairment on the LOS. We are currently working on a separate manuscript that looks at that very topic. However, for the reasons given above related to sample size and also due to the fact the present manuscript focuses on balance, we would rather keep our topic specific to balance. We agree with the reviewer, though, and have included that suggestion as part of our limitations (lines 543-544). 

4. The DWI/DTI acquisition parameters can be better described for sake of replication by other groups. Furthermore, considering that many of the ROI selected for the study has crossing fibers, limitations related to inaccurate estimation should be mentioned. I also feel that not using other DTI metrics (RD, MD, AD) may not be in best interest of the data presented. If available, the combinations of FA, AD, MD, and RD can give a lot of information with respect to nature of WM injury (neurodegeneration, demyelination, etc.) which may be pertinent. This can be added as supplementary material if word economy is a concern.

Author’s response: Thank you for the suggestions. We have provided additional details on the DTI acquisition parameters in the Methods, Image Acquisition and Processing section (lines 165-172 & 195-197). We have also mentioned the inaccurate estimation as a limitation of our study (lines 551-554). Regarding the use of DTI metrics other than FA, that is a valid point. When planning our study, we debated about reporting other DTI metrics. However, FA is arguably the most sensitive and commonly used marker of white matter integrity in TBI, and including additional DTI metrics would have generated a larger number of comparisons that would need to be corrected for, which would have washed out any potential findings. Therefore, we have computed those metrics (RD, MD, AD) but opted to analyze them without correcting for multiple comparisons and present them as supplementary information in Appendix S2 (Tables 6-11). This justification was added to the manuscript under the Methods section lines 184-193, Results section lines 304-307, 328-330 and Discussion section/Limitations lines 561-565. Also, two new citations were added (61 and 62) to justify our focus on FA (line 562).

5. The discussion could be made more informative if the injured tracts and the cortical/subcortical structures it connects is given more representation. Different components of LOS and SOT and their significance can also be described better.

Overall, this is a good study and worth reading. Thank you!

Author’s response: Thank you for your suggestions. Given that this is an exploratory study, we have discussed the injured tracts and the regions they connect as well as the functions they are involved in performing in a manner that succinctly addresses why disruptions in communication between these regions may lead to impairment on the implicated aspects of the SOT and LOS tasks. We strongly agree that these relationships and their respective cognitive functions deserve more in-depth attention in future studies that can focus more specifically on specific tracts and their associated brain regions. The intention of our study was to initially identify tracts that are likely involved in performance of these tasks and which, when injured, can affect performance on them. Our intention would be for this work to stimulate future studies that can explore these tracts and their related brain structures more in depth, focusing on specific tracts rather than a broad overview as was conducted in our study, to give other groups a general road map of the tracts that may be worth investigating further. 

Regarding your suggestion about different components of LOS and SOT and their significance being better described we responded to this suggestion when we addressed a similar question by the other reviewer. As the other reviewer suggested that we correlate SOT and LOS, we performed the suggested analysis and included information about those results in the discussion (lines 457-466). 

Reviewer #2: Overview:

This study analyzed the associations between the SOT/ LOS and FA of the brain WM areas in 56 participants with TBI. In addition to replicate a publicated study about the SOT and WM relationship, this study also reported significant correlations between abnormal LOS scores and impaired WM integrity in the cingulum, corpus callosum, corticopontine tracts, fronto-occipital fasciculi, longitudinal fasciculi and optic tract.

This study is of scientific interesting and may improve understandings on TBI-induced balance disturbances. However, results and discussion sections are somewhere redundant and less focusing. Major revisions maybe needed.

Major concerns:

(1) In Materials and methods: What atlas was used for ROI selection? and how to transfer the ROI-masks onto individual space? Please provide citations. Since the second objective is exploratory in nature, why not analyze all white matter ROIs in atlas to find out which specific white matter tracts are significantly correlated with LOS? 

Author’s response: ROIs were automatically defined in subject space using the DOTS tract segmentation algorithm, as cited (citation 23) in the original submission (lines 194-195). In the revised submission, we have provided additional details about the algorithm in the Methods, Image Acquisition and Processing section (lines 195-197). Thank you for the suggestion to analyze all white matter ROIs for potential correlations with the LOS test. There are two reasons why we chose a hypothesis-driven DTI approach with selected tracts when it comes to LOS: 1) we wanted to be able to build a parallel between the SOT and LOS results, so it made more sense to explore the same ROIs, 2) the inclusion of all white matter ROIs would generate a very large number of comparisons and our sample size was too small to afford a large number of corrections for multiple comparisons.

(2) In Materials and methods: It seems that the Benjamini-Hochberg multiple comparison is not used for correlation test between NeuroCom and FA, if I understood correctly. It is necessary to apply multiple comparison-adjusted p onto all correlation tests in Table 3 and 4.

Author’s response: Thank you for your comment on this. We are sorry this information was not clear enough in the manuscript and did not allow your correct understanding. We indeed applied Benjamini-Hochberg multiple comparison correction to all correlation tests in Tables 3 and 4 in the original manuscript. As indicated in the Statistical Analysis section the Benjamini-Hochberg method was used for all correlations. Also, as shown in the note of Table 3 in the original manuscript, a corrected p-value less than or equal to 0.013 was considered as significant for correlations between SOT and FA. Also, a corrected p-value less than or equal to 0.023 was regarded as significant for correlations between LOS and FA (Table 4). Nevertheless, to make it even clearer that we used the Benjamini-Hochberg method to correct for multiple comparisons in all correlation analyzes, we edited the Methods & Results section in multiple instances to reflect that (methods/statistical analysis section: lines 224-225; results/SOT and DTI FA associations: lines 292-293; results/ LOS and DTI FA associations: lines 316-317; results/SOT and LOS associations: lines 366-367; Table notes for Tables 3-5). 

(3) In Results: The first correlation test between SOT and FA is designed to replicate a previous study by Caeyenbergh et al. (ref#18). However, since the current data set has different features from Caeyenbergh’s study, in terms of age, inclusion of healthy controls, and sample size, it is not surprising to observe inconsistent findings. It would be clearer to deemphasize the discussion of the comparisons with ref#18, and focusing on the interpretations of significant relationship and neuroanatomy. 

Author’s response: Thank you for this observation. Because one of our main objectives was to replicate Caeyenbergh’s study, we feel strongly compelled to highlight how our results compare to theirs in the discussion. We agree with the reviewer that it is not surprising to observe inconsistencies given the differences in the dataset. We believe we have addressed a that thoroughly in the original manuscript (lines 433-452). Nevertheless, we added another sentence to the discussion to state our observation of inconsistent findings were not surprising (lines 408-410). 

(4) In Results: In this study, the exploratory LOS performed much better than commonly accepted SOT in differentiating TBI from controls. although the authors addressed that they used the manufactory provided normative data because they don't have actual healthy controls, one may wonder whether the significant group difference could have been resulted from the substantially different feature of the normative dataset that provided by NeuroCom. Additional tests may need to help understanding whether LOS is superior to SOT in describing postural control in TBI. For example, correlation tests between SOT and LOS subcategories may help to identify the similar or disassociated measurement among these tests.

Author’s response: Thank you for this question. In the introduction of the paper we tried to establish that neither test is superior to the other but that they are complementary because they measure different aspects of postural control. You raise a valid point, though, about associations between the tests. We accepted your suggestion and added a correlation analysis between the LOS and SOT as a third objective of the study (lines 72-73). We also edited our statistical analysis section accordingly (lines 221-222), created a new table (Table 5) with the new results which are now presented in lines 365-376, and finally discussed those associations in lines 456-462 of the Discussion section. Also, a new citation was added (citation 36) to support the discussion of those results in line 462.

(5) In Results: Table 3 and 4 showed clearly asymmetric findings. The asymmetric findings are common in TBI but the current study has no comments on these findings. Since this manuscript is focused on symmetric postural test and white matter tracts, it is better to perform the analysis in averaged bilateral tracts, or keep the current findings and adding with a careful discussion of asymmetry.

Author’s response: Thanks for the great suggestion. We agree and have now performed the analysis in averaged bilateral tracts and generated new results. The manuscript was edited to reflect that change in the following sections: Abstract (lines 11, 13-15), Methods/Regions of interest (ROIs) line 209, and Methods/Statistical Analysis lines 220-221. The new results have been written into the the manuscript accordingly, as the entire Tables 3 and 4 have been redone, and the following sections have been changed: Results, lines 293-304, lines 311-314, lines 321-328, lines 333-335, lines 341,345 and 348, and Discussion line 390, 454, 480-481, 492 and 518. Additionally, Figure 3 has been redone and the left corticospinal tract data that was in the left plot of the original Figure was replaced by the superior longitudinal fasciculus data in the new Figure. The text of the manuscript that relates to the new Figure 3 was also edited (lines 355-356, and 362-363). 

Minor concerns:

(6) In Materials and methods: Please provide demographic details of NeuroCom age-referenced normative data set, e.g., number of controls, mean (SD) age and sex.

Author’s response: We thank the reviewer for this suggestion and we have added that information to the text of the manuscript under Methods section/Balance testing in lines 152-162.

(7) In Materials and methods: Please be sure that No. of directions at b=300 is 10, and No. of directions at b=1100 is 60.

Author’s response: Thank you for your concern. This was indeed the diffusion acquisition protocol. It was designed in this manner for legacy reasons. 

(8) Typo on page#4: "paediatric" should be "pediatric".

Author’s response: That correction had been made. Thank you.

---

## [Decision Letter · Decision Letter 1]

4 Jul 2023

Associations between white matter integrity and postural control in adults with traumatic brain injury

PONE-D-23-06025R1

Dear Dr. Zampieri,

We’re pleased to inform you that your manuscript has been judged scientifically suitable for publication and will be formally accepted for publication once it meets all outstanding technical requirements.

Kind regards,

Monika Błaszczyszyn

Academic Editor

PLOS ONE

Additional Editor Comments (optional):

Reviewers' comments:

Reviewer's Responses to Questions

**Comments to the Author**

1. If the authors have adequately addressed your comments raised in a previous round of review and you feel that this manuscript is now acceptable for publication, you may indicate that here to bypass the “Comments to the Author” section, enter your conflict of interest statement in the “Confidential to Editor” section, and submit your "Accept" recommendation.

Reviewer #1: All comments have been addressed

2. Is the manuscript technically sound, and do the data support the conclusions?

Reviewer #1: Yes

3. Has the statistical analysis been performed appropriately and rigorously? 

Reviewer #1: Yes

4. Have the authors made all data underlying the findings in their manuscript fully available?

Reviewer #1: Yes

5. Is the manuscript presented in an intelligible fashion and written in standard English?

Reviewer #1: Yes

6. Review Comments to the Author

Reviewer #1: The authors have satisfactorily addressed all of the points raised. The article is of importance and interest to both clinicians and researchers. Thank you for the opportunity to review this manuscript.

7. PLOS authors have the option to publish the peer review history of their article (what does this mean?). If published, this will include your full peer review and any attached files.

Reviewer #1: No

---

## [Editor Report · Acceptance letter]

12 Jul 2023

PONE-D-23-06025R1 

Associations between white matter integrity and postural control in adults with traumatic brain injury 

Dear Dr. Zampieri:

I'm pleased to inform you that your manuscript has been deemed suitable for publication in PLOS ONE. Congratulations! Your manuscript is now with our production department. 

Kind regards, 

on behalf of

Dr. Monika Błaszczyszyn 

Academic Editor

PLOS ONE